# High-frequency synthetic apomixis in hybrid rice

Aurore Vernet[1,2,11], Donaldo Meynard[1,2,11], Qichao Lian [3], Delphine Mieulet[1,2,4], Olivier Gibert [1,2,5,6], Matilda Bissah[7], Ronan Rivallan [1,2], Daphné Autran [4], Olivier Leblanc[4], Anne Cécile Meunier[1,2], Julien Frouin [1,2], James Taillebois[1], Kyle Shankle[8], Imtiyaz Khanday [9,10] ✉, Raphael Mercier [3] ✉, Venkatesan Sundaresan [8,9,10] ✉ & Emmanuel Guiderdoni [1,2] ✉

Introducing asexual reproduction through seeds – apomixis – into crop species could revolutionize agriculture by allowing F1 hybrids with enhanced yield and stability to be clonally propagated. Engineering synthetic apomixis has proven feasible in inbred rice through the inactivation of three genes (*MiMe*), which results in the conversion of meiosis into mitosis in a line ectopically expressing the *BABYBOOM1* (*BBM1*) parthenogenetic trigger in egg cells. However, only 10–30% of the seeds are clonal. Here, we show that synthetic apomixis can be achieved in an F1 hybrid of rice by inducing *MiMe* mutations and egg cell expression of *BBM1* in a single step. We generate hybrid plants that produce more than 95% of clonal seeds across multiple generations. Clonal apomictic plants maintain the phenotype of the F1 hybrid along successive generations. Our results demonstrate that there is no barrier to almost fully penetrant synthetic apomixis in an important crop species, rendering it compatible with use in agriculture.

Heterosis, also termed hybrid vigor, refers to the higher performance of a hybrid progeny in comparison to those of its parents. Several underlying mechanisms (dominance, overdominance, and epistasis) have been proposed to explain the genetic and molecular bases of heterosis[1]. Heterosis has been harnessed in crops, notably through the development of F1 hybrids in seed crops that exhibit superior yield potential and stability. However, F1 hybrid seeds must be renewed at every crop season because F2 progeny seeds are prone to trait segregation and lower global plant performance. Rice, the staple food for more than half of mankind, has a naturally low outcrossing rate and the production of F1 hybrid seeds relies on the implementation of complex male sterility systems[2], resulting in a high seed cost. Thus, the dissemination of hybrid rice is essentially restricted to areas with efficient and well-established seed production and distribution systems. As a result, the benefits of hybrids have not yet reached a large number of rice farmers, and the higher production potential, enhanced stability under environmental fluctuations, and lower input requirements of hybrids remain largely unexploited.

A potentially revolutionary alternative to male sterility systems is the propagation of F1 seeds over generations in an immortalized manner through an asexual, clonal mode of reproduction called apomixis. Apomixis occurs naturally in more than 120 genera of angiosperms, notably in some wild relatives of crops such as maize, wheat, and millet[3]. However, attempts to find naturally occurring apomixis in

[1]CIRAD, AGAP institute, Avenue Agropolis, 34398 Montpellier, France. [2]University of Montpellier, CIRAD-INRAE-Institut Agro-University of Montpellier, Montpellier, France. [3]Max Planck Institute for Plant Breeding Research, Carl-von-Linne Weg 10, D-50829 Cologne, Germany. [4]DIADE, University of Montpellier-IRD-CIRAD, Montpellier, France. [5]CIRAD, UMR Qualisud, Rue Jean-François Breton, 34398 Montpellier, France. [6]University of Montpellier, Avignon University, CIRAD, Institut Agro, Reunion University, Montpellier, France. [7]CSIR, Plant Genetic Resources Research Institute, P. O. Box 7 Bunso, Ghana. [8]Dept. of Plant Biology, University of California, 1 Shields Ave, Davis, CA 95616, USA. [9]Dept. of Plant Sciences, University of California, 1 Shields Ave, Davis, CA 95616, USA. [10]Innovative Genomics Institute, University of California, 2151 Berkeley Way, Berkeley, CA 94704, USA. [11]These authors contributed equally. Aurore Vernet, Donaldo Meynard. ✉e-mail: khanday@ucdavis.edu; mercier@mpipz.mpg.de; sundar@ucdavis.edu; emmanuel.guiderdoni@cirad.fr

crops or to transfer the genetically characterized apomictic loci from wild relatives to crops have so far failed[4].

Although harnessing naturally occurring apomictic mechanisms for crops has so far been unsuccessful, recently, the possibility of engineering synthetic apomixis in plants has emerged[5,6]. Synthetic apomixis recapitulates a natural mode of apomixis, which is characterized by the formation of an unrecombined and unreduced diploid egg cell followed by its parthenogenetic development, leading to a clonal diploid zygote[7]. In synthetic apomixis, meiosis is converted to mitosis through a set of three mutations (called *MiMe* for *Mitosis instead of Meiosis*[8,9]) that target the three features that differentiate meiosis from mitosis: First, recombination and pairing are suppressed through the inactivation of a member of the recombination initiation complex (e.g., SPO*11-1*[10] or *PAIR1*[11]). Second, the joint migration of sister chromatids at meiosis is replaced by their separation through the inactivation of the cohesin *REC8*[12]. In the last step, the second meiotic division is omitted through the inactivation of the cell-cycle regulator *OSD1*[13]. Using *MiMe* as a platform, three strategies have been used to induce unreduced and unrecombined egg cells to develop into diploid clonal embryos: (i) crossing *MiMe* with a *cenh3* mutant line expressing a CENH3-variant protein[14] that induces paternal genome elimination in the zygote[15]; (ii) inactivating the sperm cell-expressed phospholipase gene (*NLD/MATL/PLA1*)[16] in *MiMe*[17] that likely also contributes to paternal genome elimination[18]; or iii. expressing a parthenogenetic trigger in the egg cell[19]. The first strategy, implemented in *Arabidopsis thaliana*, requires a crossing step and is, therefore, a non-autonomous system as it cannot be propagated by self-fertilization. In the second strategy, the *NLD/MATL/PLA1* mutation was found to alter plant fertility in rice (the fertility of the *pair1/osrec8/ososd1/osmatl* quadruple mutant is 10% of that of the control *MiMe*), and the frequency of clonal seeds in the resulting apomictic plants remained low (6–8%)[17]. In the third strategy, synthetic apomictic plants were generated through the induction of the triple *MiMe* mutation by CRISPR/Cas9 in a rice line ectopically expressing the rice *BABYBOOM1* (*OsBBM1*) AP2 family transcription factor specifically in egg cells[19]. Fertile apomictic rice plants produced clonal seeds at a rate of 10–30%, a frequency that remained stable over more than seven generations[20].

Although these three studies represented important breakthroughs, the frequency of formation of synthetic apomictic seeds remained far too low to envisage the implementation of the technology in agriculture. Increasing the penetrance of the apomictic phenotype in the third strategy is therefore an important requisite. Furthermore, integration into breeding would be facilitated if F1 hybrid seeds could be engineered in a single step with an all-in-one T-DNA construct containing both the CRISPR/Cas9-mediated *MiMe* inactivation system and the parthenogenesis inducer system. Triple homozygous/biallelic T0 *MiMe* mutants would then directly form seeds containing diploid, clonal T1 embryos.

In the present report, we introduce an all-in-one T-DNA construct into the F1 seed embryo-derived calli of a commercial hybrid of rice. We observe a high frequency of clonal seeds (>80%) in the majority of the triple *MiMe* mutants and some lines attain a >95% frequency of clonal seeds, which appear to be stable over three generations. The phenotype and the genotype of the apomictic lines are similar to those of the original F1 hybrid and remain stable over generations.

## Results

### Generation and characterization of transgenic events
We selected the commercial F1 hybrid BRS-CIRAD 302 rice, which is known for its superior grain quality, to induce apomixis in a single step by combining both the inactivation system creating the triple *MiMe* mutations and the BBM1 parthenogenesis inducer. Three T-DNA constructs were prepared for introduction into BRS-CIRAD 302 F1 seed embryo-derived callus cells (Fig. 1A): the first construct (*sgMiMe* = T313) was designed to induce by CRISPR/Cas9 the simultaneous

inactivation of *PAIR1*, *REC8*, and *OSD1*, leading to suppression of meiosis (apomeiosis)[9,19]. The second construct (*sgMiMe_pAtECS:BBM1* = T314), created in the T313 background, additionally carries *BBM1* driven by the egg cell-specific promoter of the Arabidopsis *EC1.2* gene[21,22], a cassette triggering parthenogenesis from the egg cell[19]. As the level of activity of the Arabidopsis promoter in rice could be a limiting factor, possibly responsible for the as yet incomplete penetrance of synthetic apomixis, the third construct (*sgMiMe_pOsECS:BBM1* = T315) contains the *OsBBM1* gene driven by the egg cell-specific promoter of the rice ortholog (*ECA1.1*) of the Arabidopsis *EC1.2* gene. T314 and T315 were both designed to induce synthetic apomixis (Fig. 1B).

*Agrobacterium*-mediated transformation of mature F1 seed embryo-derived calli of BRS-CIRAD302 with T313, T314, and T315 T-DNAs generated 41, 49, and 88 confirmed primary (T0) transformants, respectively (Supplementary Table 1). The frequency of fertile T0 plants was on average 53%, 49%, and 44%, in T313, T314, and T315 T0 populations, respectively, with a large variation in harvested seeds ranging from 1 to close to 300. The full sterility of a significant fraction (here 50%) of the T0 events was not unexpected since inactivation of only one or two of the three *MiMe* genes leads to meiosis failure and sterility, with only the single *osd1* or the triple *MiMe* mutants retaining fertility[9,11,23].

The efficiency of CRISPR/Cas9-mediated mutation was first evaluated by examining lesions at the *OSD1* locus in all the T0 plants. The frequency of biallelic/homozygous editing was 85%, 82%, and 73% in T313, T314, and T315 T0 populations, respectively (Supplementary Table 2). There was no striking difference in the number of T1 seeds harvested from plants not edited at the *OSD1* locus and those with frameshift lesions at this locus (Supplementary Fig. 1). We then examined the mutations at *PAIR1* and *REC8* in T0 plants with more than 10 T1 seeds, which revealed that the vast majority of the events were either wild-type or harboring mutations at the two alleles of the three *MiMe* target loci (Supplementary Table 3).

### Ploidy of T1 to T3 progenies
We identified 4, 10, and 18 *MiMe* mutants harboring triple frameshift biallelic/homozygous mutations in *PAIR1*, *REC8*, and *OSD1* in the examined T313, T314, and T315 fertile T0 plant populations. *MiMe*-only mutants should produce tetraploid progeny plants resulting from the fusion of male and female diploid gametes (Fig. 1B). Indeed, flow cytometry analysis showed that all the progeny plants of the four selected *sgMiMe* (T313) events were tetraploid (*n* = 109). (Table 1 and Fig. 1C). In contrast, if parthenogenesis is triggered, a fraction of the T1 progeny should be diploid as previously shown at a frequency of 10–30%[19] (Fig. 1B). Surprisingly, 9 of the 10 *sgMiMe_pAtECS:BBM1* (T314) and 13 of the 18 *sgMiMe_pOsECS:BBM1* (T315) independent transformants tested produced more than 80% diploid progeny plants—several of them at a rate of more than 97%—suggesting a very high frequency of parthenogenesis (Table 1). The small number of other transformants produced either exclusively tetraploids or a lower frequency of diploids, suggesting a null or lower expression of the *BBM1* transgene in these events. Ploidy level as determined by flow cytometry was systematically confirmed by phenotypic observation which allows easy discrimination of diagnostic traits of tetraploid plants (e.g., awned and large spikelets, darker and thicker leaves, lower tillering) from those of diploid plants. Altogether, these results indicate an unexpectedly high frequency of induction of parthenogenetic plants from the egg cell in T0 T314 and T315 *MiMe* events.

To determine whether this high frequency remains stable across generations, we selected two T314 (15.1 and 37.7) and four T315 (3.2, 5.4, 8.1, and 8.2) events, which produced diploid progeny (T1) at a rate of more than 92%. Determination of ploidy level in T2 progeny plants (*n* > 40) of five T1 plants per line showed a very high frequency of diploids, consistent with those

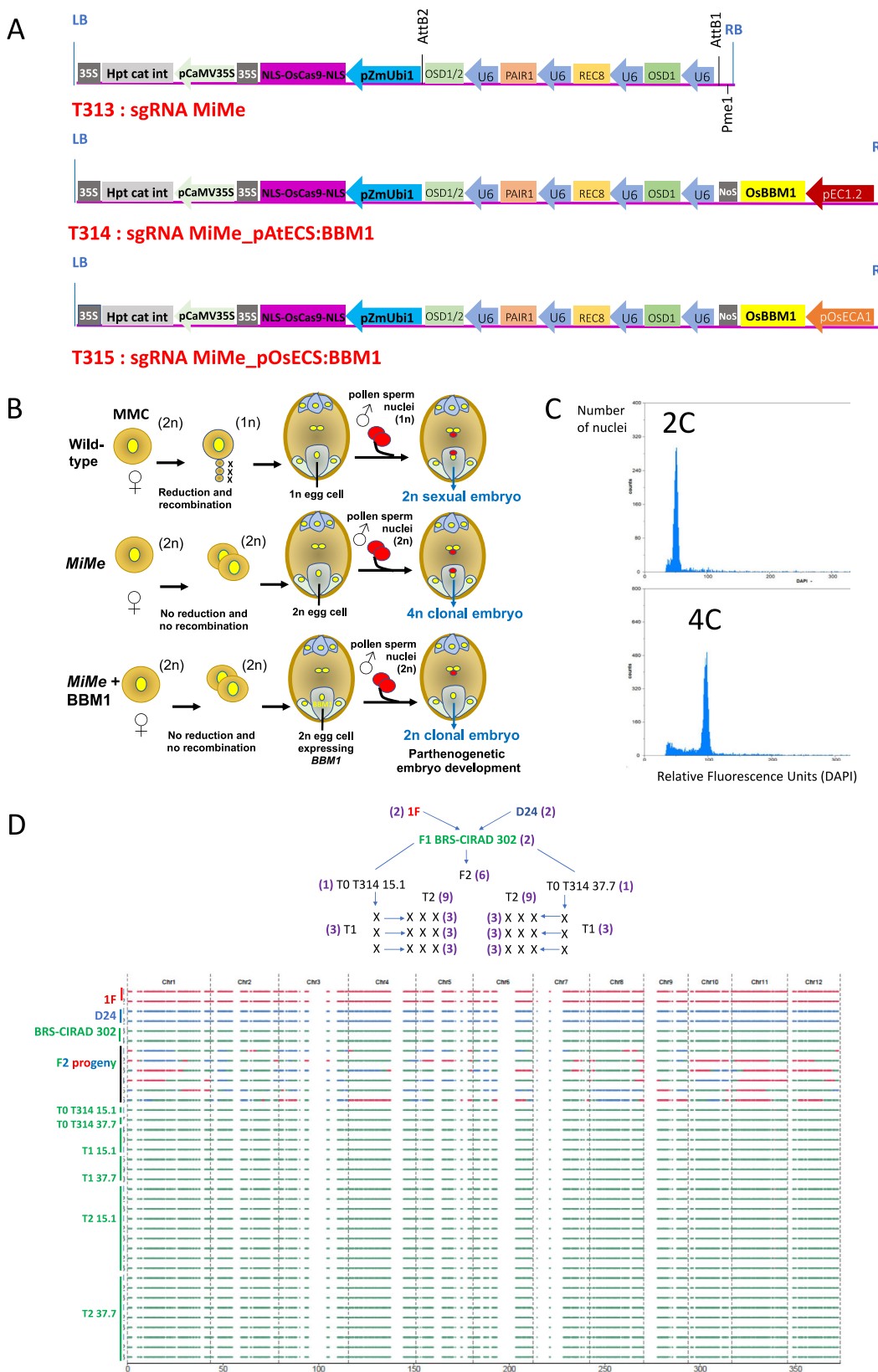

observed in T0 (Table 2) (see Supplementary Table 4 for details). We then analyzed T3 progeny plants (*n* = 100) from three T2 plants in two T314 (15.1 and 37.7) and two T315 (5.4 and 8.1) events. The germination frequency of T3 seeds was close to 100% in all four events. The four events exhibited average diploid plant formation frequencies that were higher than 90% with events

T314 15.1 and T315 5.4 exhibiting frequencies close to 99% and 97%, respectively (Table 2) (see Supplementary Table 5 for details). The high frequency of diploid plants along several generations suggests that the combination of *MiMe* with egg cell expression of *BBM1* triggers clonal reproduction through seeds with high penetrance.

**Fig. 1 | Ploidy and genotype of progeny plants of transformation events harboring the T314 and T315 apomixis-inducing T-DNA constructs. A** Schematic representation of the T-DNA constructs used to induce the triple *MiMe* mutation and the triggering of parthenogenesis, resulting in synthetic apomixis. Upper: T313 sgRNA MiMe T-DNA Middle: T314 sgRNA MiMe_pAtECS:BBM1 T-DNA Lower: T315 sgRNA_pOsECS:BBM1 T-DNA: LB and RB: left and right borders of the T-DNA; p35S: promoter of the Cauliflower Mosaic Virus (CaMV); 35S: polyadenylation signal of CaMV; hpt cat int: hygromycin phosphotransferase II with castor bean catalase intron; ZmUbi: promoter, first intron and first exon of the maize Ubiquitin 1 gene; "Os"Cas9: rice-optimized Cas9 coding sequence; NLS: nucleus localization signal fused to Cas9; OSD1, OSD1/2, PAIR1 and REC8: Four cassettes including sgRNAs (20 bp crRNA specific to the target gene + 82 bp tracR RNA) driven by the Pol III U3 promoter targeting OsOSD1, PAIR1, and OsREC8; EC1.2: egg cell-specific promoter from Arabidopsis[21,22]; OsECA1: egg cell-specific promoter from rice[31,50]. **B** Principle for formation of tetraploid and diploid clonal progenies in *MiMe* and *MiMe + BBM1* plants, respectively. **C** Representative flow cytometry histograms of DAPI-marked nuclei suspensions isolated from young leaf blade of a diploid (upper) and tetraploid (lower) progeny plants. **D** Upper: Genealogy of the plants selected for whole-genome sequencing including 1 F and D24 parents (two plants each); heterozygous F1 hybrid BRS-CIRAD 302 (two plants), six F2 sexual progenies; T314 15.1 and 37.7 primary transformants (T0); three T1 progeny plants of each of the sequenced T0 plants; three T2 progenies of each of the sequenced T1 plants (i.e., nine T2 plants per event). Lower: Graphical representation of genotypes of the 12 rice chromosomes established from whole-genome sequences of homozygous parents, heterozygous F1 hybrid, F2 progeny plants, T0 events T314 15.1 and 37.7 and their T1 and T2 progenies. Changes in color along F2 progeny chromosomes mark heterozygous-to-homozygous transitions resulting from meiotic crossovers.

**Table 1 | Ploidy of T1 progeny plants of *MiMe* BRS-CIRAD 302 primary transformants harboring the sgMiMe (T313), sgMiMe_pAtECS:BBM1 (T314), and sgMiMe_pOsECS:BBM1 (T315) T-DNAs**

| T-DNA construct | Number of T0 event | Number of T1 plants analyzed | Number of 2n plant | Number of 4n plants | % diploids |
|---|---|---|---|---|---|
| T313:sgMiMe | 4.3 | 27 | 0 | 27 | 0 |
| | 9.1 | 22 | 0 | 22 | 0 |
| | 12.1 | 20 | 0 | 20 | 0 |
| | 21 | 20 | 0 | 20 | 0 |
| T314:sgMiMe_pAtECS:BBM1 | 12.2 | 48 | 44 | 4 | 92 |
| | 15.1 | 65 | 64 | 1 | 98 |
| | 15.3 | 41 | 40 | 1 | 98 |
| | 16.1 | 37 | 30 | 7 | 81 |
| | 23.1 | 16 | 0 | 16 | 0 |
| | 37.7 | 38 | 35 | 3[a] | 92 |
| | 37.9 | 13 | 13 | 0 | 100 |
| | 44.1 | 12 | 11 | 1 | 92 |
| | 44.2 | 20 | 18 | 2 | 90 |
| | 46.2 | 26 | 21 | 5 | 81 |
| T315:sgMiMe_pOsECS:BBM1 | 1.1 | 12 | 12 | 0 | 100 |
| | 3.2 | 46 | 44 | 2 | 96 |
| | 3.3 | 59 | 51 | 8[a] | 86 |
| | 5.1 | 8 | 8 | 0 | 100 |
| | 5.4 | 44 | 41 | 3 | 93 |
| | 5.5 | 16 | 15 | 1 | 94 |
| | 6.1 | 32 | 31 | 1 | 97 |
| | 7.2 | 25 | 0 | 25 | 0 |
| | 7.4 | 16 | 1 | 15 | 6 |
| | 8.1 | 65 | 62 | 3 | 95 |
| | 8.2 | 71 | 69 | 2 | 97 |
| | 8.3 | 51 | 48 | 3[b] | 94 |
| | 16.1 | 16 | 7 | 9 | 44 |
| | 16.3 | 13 | 9 | 4 | 69 |
| | 24.1 | 31 | 30 | 1 | 97 |
| | 31.1 | 19 | 17 | 2 | 89 |
| | 31.4 | 16 | 0 | 16 | 0 |
| | 41.2 | 26 | 25 | 1 | 96 |
| | 41.6 | 20 | 19 | 1 | 95 |

The presence of the egg cell-specific promoter:BBM1 cassette was ascertained in all the T314 and T315 events. The number of plants analyzed varies according to T1 seed availability. Events T314 15.1 and 37.7 and events T315 3.2, 5.4, 8.1, and 8.2 were selected for further analysis on the basis on both the score and confidence of diploid frequency.
[a]Includes one 2n/4n chimeric plant.
[b]Includes two 2n/4n chimeric plants.

## Genotype and phenotype of diploid progenies

Clonal reproduction is expected to maintain the heterozygosity of the F1 hybrid across generations. To determine if heterozygosity was maintained in T1 and T2 progeny plants, we examined the segregation of four diagnostic markers, which are polymorphic between 1 F and D24, the two parents of BRS-CIRAD 302, and distributed on different chromosomes (Supplementary Fig. 2). These four markers segregated independently in an F2 population, as expected. In contrast, the

**Table 2 | Frequency of diploid plants in T2 and T3 progenies of selected sgMiMe_pAtECS:BBM1 (T314) and sgMiMe_pOsECS:BBM1 (T315) events**

| T-DNA constructs | Event | T1 | T2 | | T3 | |
|---|---|---|---|---|---|---|
| | | n > 40 T1 progenies (%) | Progenies of 5 T1 plants n > 40 T2 plants per progeny | | Progenies of 3 T2 plants n = 100 T3 plants per progeny | |
| | | | Average | Range(%) | Average | Range(%) |
| T314:sgMiMe_pAtECS:BBM1 | 15.1 | 98 | 96.6 | 95–100 | 98.7 | 98–99 |
| | 37.7 | 92 | 89.5 | 80–96 | 93.7 | 92–96 |
| T315:sgMiMe_pOsECS:BBM1 | 3.2 | 96 | 93.2 | 86–100 | ND | ND |
| | 5.4 | 93 | 94.8 | 92–98 | 96.7 | 96–98 |
| | 8.1 | 95 | 93.8 | 83–100 | 92.7 | 92–94 |
| | 8.2 | 97 | 88.7 | 83–95 | ND | ND |

For T2 ploidy determination, at least 40 progeny plants from 5 individual T1 plants (i.e., at least 200 plants per event) were analyzed. For T3 ploidy determination, 100 plants from 3 individual T2 plants (i.e., 300 plants per event) were analyzed. Details are provided in Supplementary Tables 4 and 5. Observations at the T1 generation are provided to facilitate interpretation of results.
*ND* not determined.

diploid T1 and T2 progenies of all the T314 and T315 events (*n* = 765 T1s and *n* = 880 T2s) exhibited the heterozygous genotype of the F1 hybrid at all four loci, strongly suggesting that they are clones. To test further whether clonal reproduction occurred, we carried out short-read, whole-genome sequencing of the 1 F and D24 parents, BRS-CIRAD 302 F1, 6 F2 progenies, 2 T0 (events 15.1 and 37.7), and 6 T1 and 18 T2 progenies (Fig. 1D). As expected, the F2 plants exhibited recombined genotypes showing heterozygous-to-homozygous transitions at cross-over positions. In contrast, all the T0, T1, and T2 plants retained the heterozygous parental genotype in the entire genome, showing that they are completely clonal. Altogether, this shows that synthetic apomixis, induced by combining *MiMe* mutations and egg cell expression of BBM1, allows very efficient clonal propagation of F1 hybrids through seeds.

To test whether the high frequency of apomixis achieved in the BRS-CIRAD 302 hybrid is a consequence of the hybrid genetic background or a function of the design of the all-in-one construct, we also transformed our all-in-one T314 and T315 constructs into the inbred Kitaake cultivar (Supplementary Table 6A). In contrast to our previous observation of ~30% apomixis frequency with the *pAtECS:BBM1* construct[19], up to ~84% of progeny from T0 transgenics raised with the T314 construct in Kitaake were found to be apomictic diploids (Supplementary Table 6B). A similarly high frequency of apomictic T1 progenies was also observed with T315 (rice egg cell promoter fusion) as well (Supplementary Table 6C). In contrast, the rice egg cell promoter fusion *pOsECS:BBM1* by itself, in the absence of the *MiMe* cassette, did not yield any parthenogenetic progeny (0% haploid out of at least 1200 T1 plants from 20 independent T0 transformants). Thus, it is clear from these results that the construct architecture, rather than genotype or number of transformants generated, is responsible for high-frequency apomixis with the all-in-one constructs.

We next explored if clonal apomicts maintain the F1 hybrid phenotype. We first examined the gross phenotype of T1 clonal plants grown along F1 hybrid and sexual F2 progenies. Whereas F2 progenies exhibited obvious disjunction for traits such as heading date, plant height, plant habit, and seed coat color traits, the clonal progenies displayed a uniform phenotype for these traits and flowered synchronously with the F1 hybrid plants (Fig. 2A). Phenotypes were more thoroughly examined in T2 progenies of five T1 plants in two T314 (15.1 and 37.7) and two T315 (5.4 and 8.1) events, grown with control BRS-CIRAD 302 plants (Fig. 2B). No significant difference in plant traits was observed across the five T2 progeny lines derived each from a T1 plant in a given event (Supplementary Table 7). Furthermore, the apomictic lines were similar to the control F1 hybrid for most traits (Table 3). The few exceptions comprised a significantly higher tiller number in the T314 37.7 event (Dunn's pairwise comparison, *p* = 0.0017) and shorter flag leaf length and narrower antepenultimate leaf width in lines 37.7

(*p* = 0.0008) and 5.4 (*p* < 0.0001) and lines 8.1 (*p* = 0.0009) and 5.4 (*p* < 0.0001), respectively (Table 3). Of note, apomictic lines exhibited significant differences between them: T315 5.4 plants were significantly taller than T314 15.1 (*p* < 0.0001) and 8.1 (*p* < 0.0001) plants, and T315 8.1 plants exhibited significantly longer panicles than T314 37.7 plants (*p* < 0.0001). These differences could be due to somaclonal variation, which is commonly observed in tissue culture-derived rice plants[24]. In a clonal mode of reproduction, such variations are likely to be fixed. Altogether, phenotypic evaluation of progenies of T314 and T315 events shows that they exhibit a generally uniform phenotype recapitulating that of the F1 hybrid.

### Grain filling in apomictic progenies

As incomplete penetrance of *MiMe* and/or failure in parthenogenetic embryo development affect fertility, a major trait to examine in apomictic lines was grain filling. We observed a rather large range of variation (20–65% filled seeds, mean = 44.4%) in grain filling among the BRS-CIRAD 302 control plants grown in our greenhouse conditions (Table 3 and Fig. 2C and D). It is not uncommon that lines with an *indica* genetic background are less fertile than *japonica* cultivars under greenhouse conditions. As hybrid plants carry the WA cytoplasmic male sterility (cms)[25], we examined pollen viability in BRS-CIRAD 302, which is expected to be close to 100% due to the sporophytic mode of restoration of WA[26,27]. In our greenhouse conditions, BRS-CIRAD 302 anthers exhibited average pollen viability of 60% (*n* = 4 plants) (Supplementary Fig. 3B and supplementary Table 8). Nevertheless, pollen viability that is reduced by half is considered sufficient for full panicle fertilization in rice, as illustrated in commercial F1 hybrids carrying the rice Hong Liang (HL) and Boro II (BT) cms systems, which have a gametophytic mode of male fertility restoration[2]. As for the T2 clonal progenies of T314 and T315 events, they also showed significantly reduced panicle fertilities, with average grain filling rates falling within a narrow 27–35.5 % range (Duncan's test, confidence interval 95%, *p* < 0.0001) (Table 3 and Fig. 2C and D). Thus, although fertilities were reduced in both BRS-CIRAD 302 control plants and apomictic progenies in greenhouse conditions, apomicts showed further reductions in fertility compared to the BRS-CIRAD 302 control plants (Table 3).

To determine the possible causes of this reduced fertility we first examined pollen formation and viability in apomictic lines: as anticipated in *MiMe* mutants of rice, dyads were formed instead of tetrads (Supplementary Fig. 3A). Mature pollen viability reached 80–90%, demonstrating that pollen viability is unlikely to be the cause of the incomplete filling (Supplementary Fig. 3B). As incomplete *osd1* mutation penetrance in Arabidopsis female meiosis results in entry into the second meiotic division, causing a reduction in female fertility, we next examined the products of female meiosis in two apomictic lines (T314 15.1 and T314 37.7). In both events, a majority of ovules formed dyads

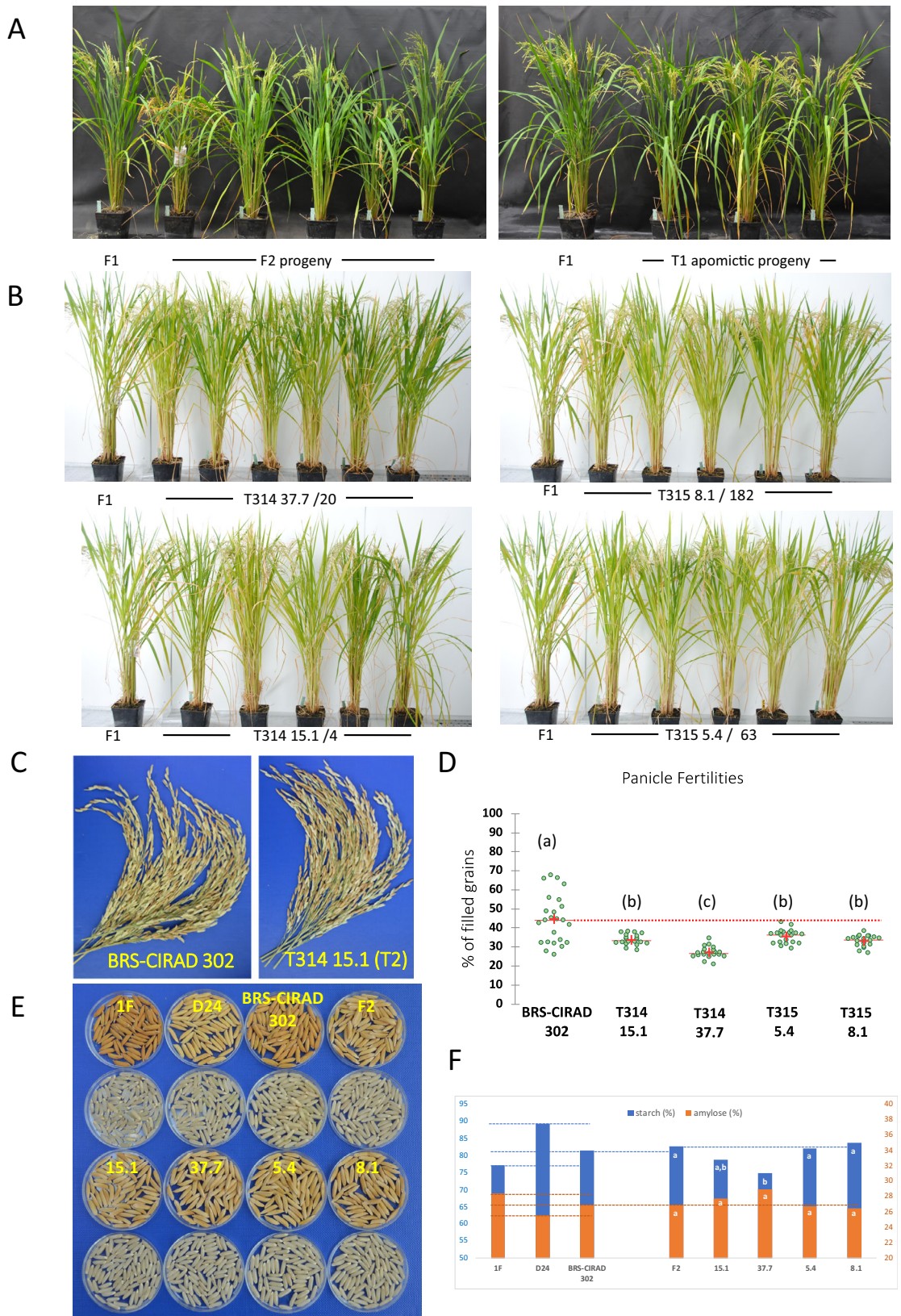

(as verified by clearing and callose deposition patterns on meiotic cell walls), followed by degradation of the micropylar spore and selection of the chalazal functional megaspore, resulting in gametogenesis (Supplementary Fig. 4A, C, D). A range of low-frequency abnormalities were also observed, such as tetrads, triads, and abortive or persisting dyads (Supplementary Fig. 4B, C), which might result in abnormal

gametogenesis that could partly explain the reduced female fertility. In addition, imbalance in chromosome segregation may occur at the first division leading to unbalanced 2n spores/gametophytes with reduced viability.

As we also produced T314 and T315 Kitaake plants with high-frequency (84%) apomixis, we next examined the seed set of T1 plants

**Fig. 2 | Phenotype, panicle fertility, and grain quality of progeny plants of selected apomictic events harboring the T314 and T315 T-DNA constructs.**
**A** Phenotypes of plants grown under controlled greenhouse conditions. Left: Five F2 progeny plants derived from the self-fertilization of BRS-CIRAD 302 compared to a BRS-CIRAD 302 F1 plant. Right: Three T1 progenies from T314 15.1 event compared to a BRS-CIRAD 302 F1 plant. **B** Phenotypes of T2 progenies grown under controlled greenhouse conditions: 5–6 T2 progeny plants of a T1 plant of events T314 15.1, T314 37.7, T315 5.4, and T315 8.1 are compared to a BRS-CIRAD 302 F1 plant. Senescent leaves of the plants have been removed for photographing.
**C** Panicles of the BRS-CIRAD 302 F1 hybrid and of T314 15.1 T2 plants. The master panicles of five distinct plants have been pooled for photographing. **D** Distribution of seed filling rate among BRS-CIRAD 302 F1 plants, and T2 progeny plants of T314

events (15.1 and 37.7) and T315 events (5.4 and 8.1 events). Average panicle fertilities of the apomictic lines represent 75.8%, 60.7, 79.9%, and 74.7%, respectively, of that of control plants (100%) (dotted red line). Significance of the differences are based on Duncan's test, dof = 4, confidence interval of 95%. **E** Husked and dehulled seeds of 1 F and D24 parents, F1 and F2 generations and apomictic lines. Upper: 1 F and D24 parents, F1 hybrid seeds harvested on 1 F parent, F2 seeds harvested on the F1 hybrid. Lower: T3 seeds harvested from apomictic plants in the four selected apomictic lines. **F** Starch and amylose content of F2 seeds and T3 seeds harvested from BRS-CIRAD 302 F1 plants and T2 apomictic plants, respectively. Different letters indicate significant differences (α-risk = 0.05) in a Kruskal–Wallis test: Event T314 37.7 exhibits a significantly lower starch content than F2 seeds and seeds of event T315 8.1 and 5.4. Source data are provided as a Source Data file.

**Table 3 | Phenotypes of T2 progenies of sgMiMe_pAtECS:BBM1 (T314) and sgMiMe_pOsECS:BBM1 (T315) events**

| Plant materials | Statistics | Plant traits | | | | | | Panicle traits | | | |
|---|---|---|---|---|---|---|---|---|---|---|---|
| | | Plant height (cm) | Number of tillers | n-1 leaf length (cm) | n-1 leaf width (cm) | Flag leaf length (cm) | Flag leaf width (cm) | Main Panicle length (cm) | Spikelets per panicle | Panicle fertility (%) | 1000-grain weight |
| BRS-CIRAD 302 | Mean | 84.8[a,b] | 13.4[a] | 50.1[a] | 1.3[a] | 31.9[c] | 1.6[a,b] | 24.5[a,b,c] | 176.4[a,b] | 44.5[a] | 21.1[a,b] |
| | SD | 2.7 | 1.8 | 4.6 | 0.19 | 6.0 | 0.13 | 1.4 | 7.9 | 3.7 | 0.2 |
| T314 15.1 | Mean | 82.8[a] | 14.4[a,b] | 48.5[a] | 1.4[a,b] | 28.2[a,b,c] | 1.6[a,b] | 24.3[a,b] | 184.9[a,b] | 33.8[b] | 20.8[b] |
| | SD | 3.9 | 2.5 | 7.1 | 0.2 | 3.7 | 0.1 | 1.5 | 11.1 | 2.0 | 0.3 |
| T314 37.7 | Mean | 85.0[a,b] | 15.2[b] | 46.0[a] | 1.4[a,b] | 26.9[a,b] | 1.5[a] | 23.6[a] | 161.9[b] | 27.0[c] | 21.2[a,b] |
| | SD | 3.3 | 1.7 | 5.1 | 0.13 | 3.7 | 0.1 | 1.0 | 8.2 | 1.4 | 0.4 |
| T315 5.4 | Mean | 87.5[b] | 14.2[a,b] | 46.4[a] | 1.6[c] | 25.9[a] | 1.6[b] | 24.7[b,c] | 208.8[a] | 35.5[b] | 22.2[a] |
| | SD | 3.3 | 1.9 | 4.1 | 0.1 | 3.0 | 0.1 | 0.9 | 12.6 | 2.4 | 0.3 |
| T315 8.1 | Mean | 83.3[a] | 13.8[a] | 48.4[a] | 1.5[b,c] | 28.6[b,c] | 1.6[b] | 25.4[c] | 204.4[a] | 33.1[b] | 21.9[a,b] |
| | SD | 4.1 | 1.6 | 4.1 | 0.1 | 2.9 | 0.09 | 1.3 | 8.6 | 2.2 | 0.2 |

At least 34 T2 plants deriving from 5 T1 plants per event were grown alongside control BRS-CIRAD 302 F1 hybrid plants (n = 15) under controlled greenhouse conditions. Details are provided in Supplementary Table 7. For all traits but panicle fertility, different letters (a,b,c) point significant differences using a non-parametric Kruskal–Wallis test with an α-risk of 0.05, followed by a pairwise comparison according to Dunn's test. For panicle fertility, different letters (a,b,c) point significant differences following a Duncan's multiple range test with a 95% confidence interval. Source data are provided as a Source Data file.
*SD* standard deviation.

from these lines. While wild-type Kitaake plants exhibited nearly 90% fertility, panicle fertility of apomictic lines reached 74% in a small number of transgenic lines examined (Supplementary Table 6B and C). Thus, although apomicts exhibited roughly 16% less fertility compared to wild-type, events with high-frequency apomixis and reasonably high (but not yet complete) fertility can be generated in a different genetic background.

### Ploidy of developing seed endosperm in apomictic lines

*MiMe* endosperm is expected to be initially hexaploid instead of triploid as it results from the fusion of the two diploid central cell nuclei and a diploid pollen sperm nucleus. To determine whether this may have an impact on grain quality in apomictic seeds, we examined the ploidy level of milky endosperm cell nuclei in developing seeds. As expected, the initial ploidy of control endosperm cells was 3n, whereas in *MiMe*-derived endosperm this initial 3 C peak was replaced by a 6 C peak (Supplementary Fig. 5). We also observed 6 C and 12 C peaks in the control endosperm, thereby confirming the well-reported phenomenon of endoreduplication occurring during the development of the cereal endosperm, which tolerates a mainly 12C–24 C level at seed maturity[28,29]. Because of this active natural endoreplication operating in the endosperm, we had anticipated that there would not be a major impact of the initial ploidy shift on grain quality.

### Seed shape and seed quality in apomictic lines

Seed shape and size are important traits contributing to grain quality and end-user acceptance. We, therefore, investigated the variation for these traits in apomictic lines with regard to grains harvested from the F1 hybrid (Fig. 2E): In sexual reproduction, the F2 grain shape is determined by the spikelet shape of the mother F1 plant. If

reproduction is clonal, then this initial shape should be preserved along generations. No significant differences were observed between F2 and T3 seeds, with the exception of a lower grain width found in line 37.7 (Dunn's pairwise comparison, p = 0.001) (Supplementary Fig. 6 and Supplementary Table 9). The thousand-grain weights of apomictic lines were not significantly different from those of F2 seeds harvested from F1 hybrid plants, either (Table 3). We then analyzed the starch and amylose contents of T3 seeds of two T314 and two T315 events compared to those of F2 seeds collected from F1 hybrid plants, grown under the same conditions as the apomictic events (Fig. 2F). The embryo and endosperm of seeds borne on the F1 hybrid have a segregating, F2 genotype whereas those of apomictic seeds share the unique F1 genotype. BRS-CIRAD 302 F1 seeds exhibit intermediary starch and amylose contents when compared to the 1 F and D24 parental lines. The F2 seeds harvested on F1 hybrid plants exhibited starch and amylose contents similar to those of F1 seeds. No significant difference was observed for starch and amylose contents between the sexual F2 and clonal T3 seeds of three of the four events. However, seeds of line 37.7 exhibited significantly lower starch content than F2 seeds (Dunn's pairwise comparison, p = 0.003) and seeds of lines 8.1 (p < 0.0001) and 5.4 (p = 0.002). Taken together, these results indicate that endosperms of clonal seeds harvested on an apomictic hybrid plant display overall identical shape and unaltered functional quality traits compared to sexual F2 seeds harvested from the F1 hybrid.

### Discussion

We report here that high-frequency (>95% over three generations) synthetic apomixis can be achieved in hybrid rice: A previous report using the same strategy, but in a two-step procedure, for inducing apomixis in the inbred cultivar Kitaake, reached a maximum efficiency

of 29%[19]. We have shown here that high-frequency apomictic events can also be generated in cv Kitaake, with a limited transformation effort. This indicates that the difference in efficiency arises from the new construct architecture rather than from the number of analyzed transformation events or the use of a specific genetic background. Whole-genome sequencing of T0, T1, and T2 generations confirmed a faithful transmission of the heterozygous genome through the synthetic apomixis mode of reproduction. Our study establishes that there is no intrinsic barrier preventing a very high level of synthetic apomixis in a crop species, stable over generations, which is a requisite for application in agriculture.

A related and more specific conclusion is that the requirement for a fertilized endosperm for viable seeds does not set an upper limit for the frequency of parthenogenesis. The necessity for an endosperm could have prevented high-efficiency clonal seed formation because of a potential conflict between parthenogenetic zygote formation and central cell fertilization that triggers endosperm development. The fertilization of the central cell requires the Egg Cell 1 (EC1) protein, which is secreted exclusively by the egg cell and necessary for inducing the two pollen-delivered sperm cells to fuse with both the egg cell and the central cell[22]. As expression of EC1 genes becomes undetectable in late stage zygotes, as shown by promoter fusions in Arabidopsis[22] and RNA-seq data from Arabidopsis[30] and rice[31], efficient initiation of parthenogenesis could lead to frequent failure of central cell fertilization and jeopardize the formation of viable seeds, setting an upper limit on apomixis frequency. Possible explanations for the very high apomixis efficiency (90–95%) obtained here are that sufficient EC1 protein persists even after initiation of parthenogenesis to activate the sperm cell for fusion with the central cell, or alternately, that the central cell—but not the egg cell—is fertilized before the egg cell develops into an embryo. In any case, our results eliminate a major concern regarding the practical applicability of this system. Mutants with the development of endosperm in the absence of fertilization have been described in Arabidopsis and rice[32,33], but these endosperms abort due to the deviation from the essential 1:2 paternal to maternal genomes ratio[32]. As shown in this study, agronomically useful efficiencies of apomixis can be achieved without incorporating autonomous endosperm development, simplifying its implementation in crop plants.

We also demonstrate that synthetic apomixis allows faithful reproduction of the F1 phenotype in apomictic lines. We observed no major differences in phenological, morphological, and grain quality traits between apomictic lines and control F1 hybrid plants, although some lines exhibited slight differences that can be likely ascribed to a somaclonal source of variation that was fixed through the apomictic mode of reproduction. Generating several apomictic lines may allow an easy selection of lines that are essentially identical to the control hybrid, as illustrated by line T314 15.1 in our study. The genetic basis of heterosis has been intensely studied, and epigenetic mechanisms have been proposed as one of the important contributions to hybrid vigor[34,35]. A study in *Hieracium*, which is a natural apomict, used recessive apomixis-deficient mutants to make apomictic hybrids[36]. These hybrids retained hybrid vigor after asexual propagation, showing that heterosis is heritable through asexual propagation. However, as *Hieracium* is a facultative apomictic species, its epigenetic control of gene expression can be expected to be adapted for asexual reproduction. In contrast, cultivated Asian rice has reproduced sexually since its domestication ~10 kya, and likely since the origin of the genus *Oryza* ~10 Mya. Studies in the obligate sexual species *Arabidopsis* have reported that hybrids are sensitive to the epigenetic state of the parents, including parent-of-origin effects[34,37,38]. Given the evidence for parental epigenetic contributions to heterosis, the absence of any paternal genome contributions in the clonal propagation of hybrids through parthenogenesis might lead to the loss of heterosis.

Nevertheless, our observations that hybrid traits are retained for two consecutive generations of clones without obvious negative consequences suggest that heterosis might be predominantly genetically controlled, and demonstrate that epigenetic factors do not restrict the utility of synthetic apomixis for rice hybrids, and possibly for other sexually reproducing crop plants of agricultural significance. Trueness-to-type and hybrid vigor should ultimately be evaluated under a field trial to confirm greenhouse findings.

The BRS-CIRAD 302 control plants exhibited incomplete and highly variable panicle fertility that ranged from 25 to 65% under greenhouse conditions. Pollen viability in the hybrid, which was on average 60%, can probably not explain the defective seed setting rate in the F1 hybrid plants. Also, in the apomictic lines, higher pollen viability (80–90%) was observed without being associated with an improved seed set. A tentative cause of the reduced panicle fertility of BRS-CIRAD 302 could be a detrimental interaction of the greenhouse environment with cytoplasmic or nuclear factors, possibly related to the WA cms system, that subsequently affected fertilization. The apomictic lines were not fully fertile either under our greenhouse conditions and exhibited an average grain filling rate ranging from 27 to 35% across the lines, representing 60–80% of that of the control F1 hybrid. To determine whether the observed variation can be ascribed to interactions between the genotype and the greenhouse environment, fertility will need to be re-examined in apomictic plants generated from manually produced F1 seeds of hybrid combinations known to exhibit full panicle fertility under greenhouse conditions. The identification of Kitaake T314 and T315 events with both high-frequency apomixis and reasonably high fertility is very encouraging in this respect.

However, as fertility seems to be consistently and significantly reduced in the apomictic lines as compared to control untransformed plants, an alternative/complementary explanation could be an incomplete penetrance of *MiMe* in female meiosis. Indeed, in both rice and Arabidopsis *osd1* mutants, a proportion (10–15%) of the female meiocytes still enter meiosis II[9,13], which is catastrophic in the *pair1 rec8* context as massive chromosome mis-segregation occurs at meiosis II. We observed here that the majority of products of female meiosis in apomictic rice lines were dyads, although tetrads and triads also occurred at low frequency, suggesting that entry into meiosis II may occur occasionally, as has been found in Arabidopsis[13] and rice[9] *osd1*. If incomplete penetrance of *osd1* reduces the fertility of the apomictic lines, a more robust prevention of meiosis II entry may be obtained by combining manipulation of *OSD1* with an additional meiotic cell-cycle regulator such as TAM[39] or TDM[40]. We also cannot exclude that post-meiotic defects, such as developmental failure of the embryo or the endosperm, also contribute to reduced fertility. The fertility and stability of synthetic apomixis will ultimately have to be tested under a range of environmental situations before a wide application in agriculture.

Parthenogenesis by the expression of BBM-related genes from apomicts was previously demonstrated in multiple cereals[41,42]. The possibility that zygote-expressed BBM genes from sexual plants can be similarly utilized to bypass fertilization has been proposed[31] and subsequently demonstrated[19]. The MiMe system to make unreduced gametes was originally developed in Arabidopsis[8]. Thus, the results from this study could have wider applicability to efforts to engineer apomixis in other crop plants. As previously discussed by several authors[4,5,7], engineered synthetic apomixis can provide a low-cost, immortalized source of F1 seeds that will allow their use by smallholder farmers. In addition, synthetic apomixis holds several specific advantages compared to male sterility-based F1 hybrid seed production. The first is a potential improvement in grain quality, which has long been a limiting factor in F1 hybrids; this has restricted their adoption and has been partly ascribed to the segregating F2 genotype of harvested seed endosperms. In synthetic apomicts, all the endosperms share the same

F1 genotype, and this should result in more predictable grain quality features. A second advantage is, for cms-based systems, avoiding the use of a single cytoplasm (e.g., WA cytoplasm) over large acreages of cultivation, which may make F1 hybrids more susceptible to disease outbreak[43] or, for environmental male sterility seed production systems, a reliance on very specific environments. A third advantage is that it could widen the breadth of tested hybrid combinations that have so far been restricted by the long and tedious preliminary process of converting the parental lines to cms prior to multisite evaluation. Progress in understanding and harnessing the dispensable genome, which has been demonstrated to harbor a wealth of adaptation genes[44], may allow a more informed choice of genome combinations for developing future climate-smart apomictic hybrids. Having an efficient tool for converting hybrids to apomixis will be very valuable in this respect. Beyond the well-reported yield performance and stability qualities, apomixis should therefore allow breeders to harness the potential of F1 hybrids that exhibit biotic and abiotic stress tolerance and are thus better equipped to deal with the challenges posed by global climate change and increased food demand.

## Methods

### Plant material and growth conditions

Commercial F1 seeds of BRS-CIRAD 302 (a CIRAD-EMBRAPA hybrid of rice released in 2010 in Brazil), were used in this study. BRS-CIRAD 302 is a high-yielding *indica/indica* F1 hybrid resulting from crossing a male sterile line of 1 F (CIRAD 464), bearing the Wild Abortive (WA) cytoplasmic male sterility, and the D24 line carrying the restorer nuclear genes. Grains harvested on BRS-CIRAD 302 exhibit superior quality, according to Graham's[45] classification, i.e., a long and slender shape, low breakage rate, and high amylose content. For ascertaining purity in hybrid seed production, 1 F harbors a recessive, brown-colored seed coat trait while D24 harbors a dominant, straw-colored seed coat trait. Seed husks borne by the F1 hybrid are therefore straw in color. Transgenic plants (T0, T1, T2, and T3 generations), parental lines 1 F and D24, BRS-CIRAD 302 hybrid, and F2 progenies were grown in containment greenhouse facilities under natural light supplemented by light provided by LEDs (12 h:12 h photoperiod) under 60% hygrometry and 28 °C day and 20 °C night temperatures. Wild-type and transgenic events (T0 and T1 plants) of cv. Kitaake were grown in UC Davis greenhouses under natural light conditions.

### T-DNA construct preparation

The 4sgMiMe (T313) construct was created by inserting a 2067 bp attB1-Attb2 fragment from the T-DNA of the pCAMBIA2300-MiMe 4sgRs vector[19], containing four single guide RNA (sgRNA) cassettes each driven by an *OsU6* promoter[46], into a pDONR207 vector by Gateway (Clontech) cloning. Two guide RNAs (5′-GAGAAATTCCGGCGGTAGGG-3′ and 5′-GCGCTCGCCGACCCCTCGGG-3′) were used for *OSD1* and one each for *PAIR1* (5′-TCGACGA-CAACCTCCTCACC-3′) and *REC8* (5′-GTGTGGCGATCGTGTACGAG-3′) (Supplementary Table 10). The fragment was then transferred into a pENTRY vector and eventually into the pCAMBIA 1300-based (www.cambia.org) 2G9 binary plasmid through an LR reaction. The 2G9 binary vector T-DNA region harbors a rice codon-optimized Cas9 coding region[47] driven by the maize ubiquitin promoter[48], a selectable cassette containing the *hpt* gene with a catalase intron[49], driven by the CaMV 35 S promoter, and a cmR ccdb Gateway cloning site.

The sgMiMe_pAtECS:BBM1 (T314) vector was prepared following blunt-end insertion of the 3088 bp *EcoR*I fragment of pGEM-Te-DD45:BBM1:Nos[19] containing the *Arabidopsis* EC1.2 egg cell-specific promoter driving the *OsBABYBOOM1* (*OsBBM1*) coding sequence terminated by the Nopaline synthase (Nos) polyadenylation signal, into the compatible *Pme*I site of T313, situated proximal to the right border of the T-DNA.

The sgMiMe_pOsECS:BBM1 (T315) vector was prepared by releasing the 2828 bp XbaI-AseI fragment of the pCAMBIA1300_pO-sECA1.1:OsBBM1:nosT plasmid, containing the promoter of the rice egg cell-specific *OsECA1* (LOC_Os03g18530) gene amplified with REG1 F and REG1 R primers (Supplementary Table 11)[31,50] driving *OsBBM1* terminated by the Nos polyadenylation signal, followed by its blunt-end insertion into the *Pme*I cloning site of T313. The inward orientation of the egg cell-specific promoter:OsBBM1 cassettes adjacent to the T-DNA RB was ascertained in both T314 and T315 by sequencing. The sequences of the sgRNAs targeting rice *PAIR1*, *REC8* and *OSD1*, the *BBM1* coding region, and the *Arabidopsis thaliana* EC1.2 promoter region sequences were those originally described elsewhere[21,22]. The binary vectors were introduced into *Agrobacterium tumefaciens* strain EHA105 by electroporation. The three constructs were cloned by the GENSCRIPT company (Leiden, the Netherlands).

### Plant transformation

Transformation of BRS-CIRAD 302 was carried out by co-cultivation of mature F1 seed embryo-derived secondary calluses with EHA105 *Agrobacterium* cell suspensions, selection of transformed cell lines based on hygromycin tolerance, and primary transformant (T0) regeneration following the procedure detailed by Sallaud and co-workers[51]. Slight changes to the procedure include the reduction of *Agrobacterium* cell suspension OD to 0.01 and lengthening of some phases of selection due to slower growth of transformed *indica* cell lines compared to those of standard *japonica* cultivars. Two rounds of transformation were carried out for the T314 and T315 vectors. Transformation efficiencies using the T313, T314, and T315 vectors in BRS-CIRAD 302 are shown in supplementary Table 1. Eventually, 41, 49, and 88 primary transformants of the 3 respective populations were transferred to the containment greenhouse and grown until the harvesting of T1 seeds. The primary transformants are numbered after both the co-cultivated callus number (e.g., callus 21) and the hygromycin-resistant cell line number (e.g., hygromycin-resistant cell lines 21.1, 21.2, and 21.3) they originate from. The several hygromycin-resistant cell lines deriving from a single co-cultivated callus are generally independent transformation events[51]. Twenty primary transformants for each of the T314 and T315 T-DNA vectors were raised in the Kitaake cultivar using *Agrobacterium*-mediated transformation with the EHA105 strain.

### Molecular characterization of primary transformants and progeny

DNA from primary transformants and T1 and T2 progeny, as well as of control plants, was isolated using MATAB and automated procedures in 96-well dishes. Quantification of T-DNA copy number in primary transformants was achieved by Q-PCR analysis of the *hpt* gene copies[52]. The presence of both Cas9 and of the egg cell-specific cassettes was ascertained by PCR in T0 transformants selected for analysis (Supplementary Table 11 for primers). The efficiency of the sgRNA at generating mutations at their respective target sites was evaluated through PCR and sequencing, using primers amplifying their target regions in the two alleles of BRS-CIRAD 302 (Supplementary Table 11). For *OSD1*, natural polymorphisms existing in the two parental sequences compared to the *Osd1-1* crRNA target sequence, originally designed for the *japonica* cv. Kitaake, made the *Osd1-1* sgRNA inefficient at inducing lesions at this site. A second target site in *OsOSD1* was originally added because of the rather low efficiency of *Osd1-1* at creating lesions in cv. Kitaake (see supplementary Table 6A). All the T1 and T2 progeny plants for which ploidy was determined were also analyzed, along with parental and hybrid and F2 control plants, for segregation of four polymorphic microsatellites markers located in the middle of the long arms of chromosomes 1 (RM1), 8 (RM25), 9 (RM215), and 11 (RM287) (Supplementary Table 11). The F2 progenies ($n = 12$) were analyzed to

check for the free segregation of the parental alleles. The SSR work was carried out at the genotyping facility of Cirad in Montpellier (France). Whole-genome sequencing of 1 F, D24, BRS-CIRAD 302 (2 plants each), 6 F2 plants, 2 T0 plants (T314 15.1 and 37.7), 6 T1 plants, and 18 T2 plants was carried out on the Illumina sequencing platform of the MPIPZ in Cologne (Germany). The genealogy of the materials used for sequencing is summarized in Fig. 1D.

## Whole-genome sequencing and genotype calling
The reference rice genome (Nipponbare/*japonica*) was downloaded from the Phytozome database[53,54]. The paired-end raw reads were first quality-evaluated using FastQC v0.11.9 (http://www.bioinformatics.babraham.ac.uk/projects/fastqc/)[55] and then mapped to the reference genome using BWA v0.7.15-r1140 with default parameters. Tandem Repeats Finder v4.09[56] was used with default parameters to scan the genome tandem repetition of DNA sequences.

To obtain high-confidence SNP markers that can differentiate the two parental alleles, we used the analysis strategy described below. First, for parental and F1 samples (with a mean depth of 14.3×), SNPs and SVs were predicted from whole-genome resequencing datasets using inGAP-family[57]. Then, potential false-positive SNPs that come from sequencing errors, small indels, tandem repeats and SVs were identified and filtered by inGAP-family[57,58]. Furthermore, those SNPs only present in one of the parents (each with two replicates, homozygous genotype), and with a heterozygous genotype in F1 plants (two replicates, reference allelic ratio larger than 0.3 and less than 0.7) and sequencing depths larger than three, were kept as high-confidence markers. In the end, we obtained a set of 267,753 high-confidence SNP markers for subsequent analysis.

For F2, T0, T1, and T2 samples (with a mean depth of 1.9×), the read count and genotype profile for SNPs were generated by inGAP-family, and then a sliding window-based method was used to construct the genome-wide genotype landscape across samples, with a window size of 200 kb and a sliding size of 100 kb. The genotype profile was visualized by using ggplot2 v3 3.5 package in R environment.

## Flow cytometry analysis
DNA content of DAPi-stained cell nuclei isolated from developing leaf blades and seed endosperms was determined by FACS using a PARTEC cell analyzer and Sysmex CyStain® UV Ploidy 05-5001 buffer (www.sysmex.de). The method for releasing nuclei into the buffer using manual chopping of leaf blade segments with a razor blade was that described in ref. 9. For endosperm nuclei, the pear-shaped developing seed was gently separated from the lemma and palea and allowed to release its milky endosperm into 0.5 ml of buffer solution using a pipette tip. Then, the turbid suspension containing nuclei was diluted in 3 ml of buffer and vortexed before FACS analysis.

## Morphological trait phenotyping
Thirty-four to forty T2 plants derived from T1 plants (*n* = 5) of each of four T0 events (T314 15.1, 37.7 and T315 5.4 and 8.1) were grown in the containment greenhouse until maturity along with fifteen BRS-CIRAD 302 F1 control plants. All the plants flowered in a synchronous manner. The following traits were recorded at the time of harvesting: Master tiller height, tiller number, antepenultimate (n − 1) leaf blade length and width, flag leaf blade length and width, master tiller panicle length, number of spikelets per panicle (average of three master tillers), filled grain frequency (%), and one-thousand grain weight.

## Plant panicle fertility and pollen viability determinations
The panicle fertility of T1 and T2 plants as well as that of control BRS-CIRAD 302 plants grown along the transgenic plants, was determined by averaging the filled spikelet/total spikelet ratio of the three main tillers of each plant. The panicle fertilities of four T2 progeny plants of each of five T1 plants (i.e., 20 T2 plants) were analyzed per T0 event. The panicle fertility of Kitaake T1 plants of two T314 and two T315 events grown alongside wild-type plants was estimated from the master tillers of five T1 plants. The pollen viability was determined by counting at least 1500 Alexander's[59] solution -stained pollen grains released from the mature anthers of two flowers collected on four T2 plants in each of four selected T0 events (T314 15.1 and 37.7, T315 5.4 and 8.1) and in control BRS-CIRAD 302. Viable pollen appear pink-colored whereas empty, unviable pollen grains appear green-colored. Alexander staining is known to overestimate pollen viability.

## Microscopy analysis of meiotic products by clearing and callose detection by aniline blue staining
Ovaries and anthers collected at pre-anthesis stages (white or pale-yellow anthers) from independent T2 plants of apomictic events T314 15.1 and T314 37.7 (and of replicated samples collected on T3 plants of event T314 15.1) were fixed in Carnoy's or FAA fixatives and rinsed in 70% ethanol. Dissected ovaries were mounted in Hoyer's clearing medium and gently squashed by pressure on the coverslip to expose ovules. Callose detection using aniline blue staining was performed as described[60]. Observation and imaging were done on a Zeiss Axioimager Z2 microscope equipped with DIC, CFP emission filter; ×40 or ×63 (NA1,4) oil immersion lenses, and a sCMOS camera (Hamamatsu ORCA Flash V2). Images were processed using ImageJ software.

## Material for grain quality characterization
All oven-dried rice kernels were dehulled prior to the manual removal of embryos. Cargo rice seeds were then ground using a ball mill (Dangoumill 300, Prolabo, France). After determination of the moisture content of an aliquot of the fine-ground rice flours by thermogravimetry in a ventilated oven at 104 °C until constant weight was reached, samples were hermetically stored at ambient temperature until use.

## Grain morphological feature
A flatbed scanner (EPSON Expression 10,000 XL) was used to acquire 800 ppi RGB images of about 50 kernels of each paddy and cargo rice seeds. The size and shape of kernels were obtained based on digital image processing to avoid the laborious measurement of individual grains with a calliper. The major axis (length) and minor axis (width) were determined from an ellipse that best fits the projected area of the binarized image of each kernel using the open source ImageJ program. For each sample, shape (length/width), thickness (E), and volume (V) were then estimated, assuming that the width was proportional to the thickness of the spheroid grain[61].

## Starch enzymatic determination
Without removal of negligible soluble oligosaccharides, a dried starchy sample aliquot (25 mg) was gelatinized at 90 °C for 1.5 h with NaOH (1 mL 0.02 N) prior to being hydrolyzed at 50 °C for 1.5 h with α-amyloglucosidase in a pH 4.5 citrate buffer. D-glucose was then indirectly evaluated via the production of gluconate through the reduction of nicotinamide adenine dinucleotide phosphate (NADPH) using hexokinase and glucose-6-phosphate-dehydrogenase enzymes by the UV method of spectrophotometry at 340 nm (R-Biopharm, 2021). The amount of starch was expressed in percent on a dry-weight basis (g starch per 100 g dried sample), using an estimated conversion factor of 0.9 between starch and D-glucose.

## Amylose calorimetric determination
Amylose content was measured by differential scanning calorimetry with a DSC 8500 apparatus (Perkin Elmer, Norwalk, USA)

using 9–10 mg db dried sample and 40 μL of 2% (W/v) L-α-lysophosphatidylcholine solution (Sigma Chemical Co., St Louis, USA) in a hermetically sealed micropan[62]. The energy of the amylose/lyso-phospholipid complex formation of the sample (J/g db) against a pure amylose standard (Avebe, Veendam, Netherlands) was estimated in duplicate during the cooling stage from 160 to 40 °C at a rate of 10 °C/min. The amount of amylose was expressed in percentage, on dry starch weight basis (g amylose per 100 g dried starch).

### Statistical analyses

As most traits did not follow a normal distribution, the statistical significance of differences between transgenic lines and controls was determined using non-parametric Kruskal–Wallis tests and an α-risk of 0.05. Multiple pairwise comparison was performed according to Dunn's procedure with a Bonferroni-corrected significance level of the *p*-value. For panicle fertility, we used Duncan's test with a significance of 0.05%. The XLSTAT software (Addinsoft, Paris, France) was used for the statistical analyses.

### Reporting summary

Further information on research design is available in the Nature Portfolio Reporting Summary linked to this article.

## Data availability

The whole-genome resequencing data of all individuals are available in the ArrayExpress database at EMBL-EBI under accession number E-MTAB-11931. Source data are provided with this paper.

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

## Acknowledgements

We wish to thank Murielle Portefaix (INRAE) for assistance with tissue culture, Eve Lorenzini, Rémy Michel, Sémi Melliti, and Christian Chaine for greenhouse support, Dr. Frédéric Bakry and Dominique Dambier for assistance and advice in flow cytometry, Armelle Soutiras and Mathilde Singer for assistance in grain quality analyses, Matthieu Dejean for the grain morphological script using the open source ImageJ program, Pierre Mournet, for advice on the Great Regional Technical Platform (GPTR) of genotyping at Cirad. This research has been conducted on the functional analysis (AFEG), cell imaging (PHIV), and biochemical phenotyping (PPB) platforms of CIRAD. The support of Dr. Christophe Périn at the initiation of the project is acknowledged. M.B. was a recipient of a Bill and Melinda Gates AWARD grant through Agropolis Fondation. This research was supported by the France-Berkeley exchange fund grant to V.S. and E.G., the CGIAR Research Program on Rice (CRP RICE) to E.G., the Innovative Genomics Institute (IGI) grant to V.S., an Alexander von Humboldt fellowship to Q.L., and core funding from the Max Planck Society to R.M. We thank Neysan Donnelly, MPIPZ, for editing the English of the manuscript.

## Author contributions

A.V., Do.M., I.K., and E.G. conducted the experiments, De.M. and I.K. designed and prepared the rice transformation constructs, and I.K. also performed Kitaake experiments. Q.L. analyzed the whole-genome sequencing data, M.B. carried out flow cytometry analyses, O.G. conducted the grain quality analyses, R.R. conducted the SSR genotyping, D.A. and O.L. carried out the cytological observations of female meiosis, A.C.M. helped with CRISPR/Cas9-induced mutation analyses, J.F. prepared DNA samples, K.S. prepared the pCAMBIA1300-pOsECA1.1:OsBBM1:nosT plasmid, and J.T. provided plant materials and provided advice on implementation of apomixis for hybrid rice. V.S., R.M., I.K., and E.G. supervised the research. E.G. drafted the manuscript, which was edited by R.M., V.S., and I.K. All authors read, commented on, and approved the final version of the manuscript.

## Competing interests

INRAE, the former employer of R.M., holds patents on *MiMe* and its use to engineer apomixis. A provisional application for an US patent has been filed on October 3rd 2022 (# 63/412,667) by UC Davis, CIRAD and MPIPZ, with I.K., V.S., E.G., De.M., and R.M. as inventors. The remaining authors declare no competing interests.
