## [Peer Review File · Nature Communications]

Reviewers' Comments:

Reviewer #1:

Remarks to the Author:

(Review Sept 2022)

I appreciate and approve of the changes that the authors have made to improve the manuscript. The methods are sound, and while it is exciting that the rates of apomictic seed formation have been increased, the seed filling data are clearly another hurdle that must be crossed.

The conclusions drawn in the Discussion are justified, although I have two small questions:

Line 278 - "The fertilization of the central cell requires the Egg Cell 1 (EC1) protein, which is secreted exclusively by the egg cell and necessary for inducing the two pollen-delivered sperm cells to fuse with both the egg cell and the central cell³⁰. Since EC1 expression becomes undetectable in the zygote^{22,31}, efficient initiation of parthenogenesis could lead to frequent failure of central cell fertilization and jeopardize the formation of viable seeds, setting an upper limit on apomixis frequency.^d". I'm wondering why the authors haven't included this data in the manuscript? Novel EC1 dynamics would be an important contribution to the field.

Line 323 - "A tentative cause of the reduced panicle fertility of the hybrid could be a detrimental interaction of the WA cytoplasm with the greenhouse environment through pleiotropic effects affecting other traits involved in fertilization." This statement is vague, and I'm not sure why the "cytoplasm" is being singled out considering there could be other GxE explanations. Or do the authors mean "germplasm"?

(Review November 2022)

I have read the responses of the authors to the first round of reviews, and have re-read the revised manuscript. I disagree with reviewer 2 that this manuscript does not contain the novelty to be published here, and I feel that the authors have made sufficient justification and changes to respond to these comments. Importantly, the authors did add new data from an additional Kitaake genotype, whose improved apomictic traits also speak to the doubts brought up by this reviewer, not to mention demonstrate the reproducibility of their tool.

Reviewer #3:

Remarks to the Author:

All of my comments and concerns have been answered and addressed.

Reviewer #4:

Remarks to the Author:

This contribution is a valuable advance in achieving apomictic rice-- a major worldwide crop. This reviewer has no further comments.

Reply to reviewers.

We wish to thank the 3 reviewers for their positive comments on our manuscript. We also wish to express our sincere gratitude to reviewer 1 who accepted to review our responses to reviewer 2's comments.

Reviewer 1:

Line 278 - "The fertilization of the central cell requires the Egg Cell 1 (EC1) protein, which is secreted exclusively by the egg cell and necessary for inducing the two pollen-delivered sperm cells to fuse with both the egg cell and the central cell³⁰. Since EC1 expression becomes undetectable in the zygote^{22,31}, efficient initiation of parthenogenesis could lead to frequent failure of central cell fertilization and jeopardize the formation of viable seeds, setting an upper limit on apomixis frequency.d". I'm wondering why the authors haven't included this data in the manuscript? Novel EC1 dynamics would be an important contribution to the field.

Response: The EC1 dynamics mentioned here are not new, but have been previously published for Arabidopsis in refs. 22 and 30, and for rice in reference 31. We have re-written the sentence to make this point clearer: "Expression of EC1 genes becomes undetectable in in late stage zygotes, as shown by promoter fusions in Arabidopsis²² and RNA-seq data from Arabidopsis³⁰ and rice³¹."

Line 323 – "A tentative cause of the reduced panicle fertility of the hybrid could be a detrimental interaction of the WA cytoplasm with the greenhouse environment through pleiotropic effects affecting other traits involved in fertilization." This statement is vague, and I'm not sure why the "cytoplasm" is being singled out considering there could be other GxE explanations. Or do the authors mean "germplasm"?

Response : We thank the reviewer for the useful comment : We have recently compared the panicle fertility of BRS-CIRAD 302 F1 hybrid bearing the restored WA cms and that of F1 plants derived from manual hybridization between 1F (maintainer or B line) and D24 under greenhouse conditions. Though again not being fully fertile , 1F/D24 plants exhibited a panicle fertility of 67.9%, 19% higher than that of the BRS-CIRAD 302 commercial hybrid plants grown alongside (n=12, p=0.008). This could indicate that transfer of WA cms into 1F (cms or A line) has dragged cytoplasmic or nuclear factors that interact negatively with the greenhouse environment for the fertility trait. The factors can be related to WA cms or others, so the reviewer is right that "germplasm" would be more appropriate. We have therefore revised to the following sentence:

"A tentative cause of the reduced panicle fertility of BRS CIRAD 302 could be a detrimental interaction of the greenhouse environment with cytoplasmic or nuclear factors, possibly related to the WA cms system, that subsequently affected fertilization".

The different performances of the rice cytoplasmic male sterile (CMS) line A and its maintainer line B under environmental stress have recently been shown to arise from differences in RNA editing (Xiong et al (2017) Frontiers in Plant Science 8:2023)